

# Improving the success of reinforcement programs: effects of a two-week confinement in a field enclosure on the anti-predator behaviour of captive-bred European hamsters

Julie Fleitz[1,2], Manfred R. Enstipp[1], Emilie Parent[1], Jonathan Jumeau[3], Yves Handrich[1] and Mathilde L. Tissier[4]

[1] Department of Ecology, Physiology and Ethology, Université de Strasbourg, CNRS, IPHC UMR 7178, Strasbourg, France
[2] Société Cofiroute, Strasbourg, France
[3] Collectivité Européenne d'Alsace, CERISE, Strasbourg, France
[4] Department of Biological Sciences, Bishop's University, Sherbrooke, Quebec, Canada

Corresponding author
Julie Fleitz, julie.fleitz@gmail.com

## ABSTRACT

Captive breeding programs are an important pillar in biodiversity conservation, aiming to prevent the extinction of threatened species. However, the establishment of self-sustaining populations in the wild through the release of captive-bred animals is often hampered by a high mortality upon release. In this study, we investigated how a 2-week confinement period within a large field enclosure affected the anti-predator behaviour of 'naive' captive-bred hamsters and how potential modifications persisted over time. During three consecutive tests, hamsters were confronted with a moving predator model (a red fox mount, *Vulpes vulpes*) and their behaviour was filmed. After the initial round of confrontation with the predator model, one group of hamsters (field group) was released into a field enclosure protected from predators, while the other group (control) remained in their individual laboratory cages. After 2 weeks, hamsters from the field group were recaptured and individuals of both groups underwent a second confrontation test. A total of 1 month after their return from the field enclosure, field hamsters were subjected to a last confrontation test. Video analysis, investigating four behavioural variables, revealed that field hamsters significantly modified their behavioural response following the 2 weeks confinement in the enclosure, while this was not the case for control hamsters. In addition, most behavioural modifications in field hamsters persisted over 1 month, while others started to revert. We suggest that an appropriate pre-release period inside a field enclosure will enable naive (captive-bred) hamsters to develop an adequate anti-predator behaviour that will increase their immediate survival probability upon release into the wild. We believe that such measure will be of great importance for hamster conservation programs.

## INTRODUCTION

Reinforcement or 'restocking' programs are widely used in biodiversity conservation to sustain or restore declining or threatened populations (*Fischer & Lindenmayer, 2000*; *Guy, Curnoe & Banks, 2013*; *Bubac et al., 2019*). Reinforcement is defined as the intentional addition of captive-bred individuals or individuals from a stable wild population to an existing group of conspecifics, to recover endangered populations (*Zlatanova, 2016*). This strategy is largely used to increase population density, to compensate for low dispersion rates, to correct skewed sex-ratios or to improve the genetic status of small populations (*Weeks et al., 2015*).

Reinforcement programs have been implemented worldwide on a variety of taxa, ranging from invertebrates to mammals (*Soorae, 2018*). Unfortunately, their success is often limited (*Beck et al., 1994*; *Black et al., 1997*; *Fischer & Lindenmayer, 2000*; *Short, 2009*). This is mainly due to the high short-term mortality of animals released during restocking programs (*Griffin, Blumstein & Evans, 2000*; *McPhee & Silverman, 2004*; *Shier & Owings, 2007*; *Brichieri-Colombi & Moehrenschlager, 2016*; *Berger-Tal, Blumstein & Swaisgood, 2020*). Four main reasons have been proposed to explain such high post-release mortality: (1) the unfamiliarity of released animals with local conditions (*Calvete & Estrada, 2004*); (2) the high risk of starvation due to the inability of captive-bred animals to forage efficiently (*Jule, Leaver & Lea, 2008*); (3) immune deficiencies (*Abolins et al., 2017*), and (4) an alteration of the instinctive anti-predator behaviour when reared in captivity over many generations (*Miller et al., 1990*; *Fischer & Lindenmayer, 2000*; *Griffin, Blumstein & Evans, 2000*).

A number of studies have investigated possible alterations of the instinctive anti-predator behaviour in prey species as a consequence of captive breeding (*Shier & Owings, 2006*; *Carrete & Tella, 2015*; *Jolly, Webb & Phillips, 2018*). Some of these studies suggest that the effectiveness of the anti-predator behaviour of fish, birds and mammals can be improved through a pre-release treatment (*Griffin, Blumstein & Evans, 2000*; *Guy, Curnoe & Banks, 2013*; *Edwards et al., 2021*). Treatments can consist of animals experiencing environmental enrichment, a soft release or antipredator training (*i.e.*, conditioning desired behaviour) before release into the wild. The latter is characterized by a training period during which an animal is exposed to a predator model coupled with aversive stimuli, such as alarm signals (*i.e.*, conditioning; *Kleiman, 1989*; *McLean, Lundie-Jenkins & Jarman, 1996*; *Griffin, Blumstein & Evans, 2000*; *Shier & Owings, 2007*). During soft releases, individuals experience a pre-release period inside a field enclosure that mimics the environment of their future release site as closely as possible but shelters them from predation. Presumably, the latter treatment improves the ability of captive-bred individuals to recognize and avoid predators after being released and allows them to familiarize themselves with new threats (*Reading, Miller & Shepherdson, 2013*; *Resende et al., 2021*). However, the success of pre-release treatments differs between species and their efficacy has rarely been tested for solitary-living prey species, which lack the horizontal transmission of survival behaviour from conspecifics (*Tetzlaff, Sperry & DeGregorio, 2019*).

One such species is the European hamster (*Cricetus cricetus*), which, until the 1970s, was abundant across Europe and Asia (*Weinhold, 2009*; *Surov et al., 2016*). However, due to habitat fragmentation, agriculture intensification, and climate change, it is now one of the most threatened mammal species in Western Europe (*Weinhold, 2009*; *Tissier et al., 2016*) and has recently been classified as "Critically Endangered" by the IUCN (International Union for Conservation of Nature; *Banaszek et al., 2020*). In a recent study, European hamsters that were bred in captivity over 15 generations in France showed a marked aggressive response when confronted with a mobile predator (European ferret, *Mustela putorius furo*), rather than fleeing and hiding in an available shelter (*Tissier et al., 2019*). Such behavioural response is not consistent with the common assumption that prey species only display aggressive behaviour towards a predator when freezing or fleeing are not viable options (*Eilam, 2005*). Attacking an unknown predator is likely to be fatal for hamsters in the wild, questioning the appropriateness of such a response (*Tissier et al., 2019*). Non-appropriate behavioural responses to predation risk can be a major problem for reinforcement programs, especially when animals face a high predation pressure upon their release into the wild (*Moseby et al., 2011*; *La Haye et al., 2020*).

In addition to conservation measures focusing on habitat restoration (*La Haye, 2013*; *Tissier et al., 2018*, *2021*), reinforcement programs have been implemented in most western-European countries, in an effort to sustain and restore the most fragile hamster populations across Europe (the Netherlands: *La Haye et al., 2010*; Belgium: *Verbist, 2008*; and Germany: *Sander & Weinhold, 2008*). In France, a reinforcement program has been in place since 2002 but its success has varied considerably (*Villemey et al., 2013*; *Chaigne et al., 2015*). As part of this program, captive-bred hamsters have been released at unharvested agricultural sites every spring for the past 20 years. To prevent attacks from terrestrial predators, these sites are protected by electric fences throughout the hamster breeding season (*Villemey et al., 2013*). While such measures have generally improved the post-release survival of hamsters, their mortality following release remained high in some years (*i.e.*, up to 91% during the first 4 months; calculated from *Virion (2017)*). This was mainly due to (1) avian predation (2) terrestrial predators overcoming fences or (3) the dispersal of released hamsters to areas with little or no vegetation cover, resulting in predation. Hence, while the reinforcement program has allowed to maintain the relict hamster population in France, it has so far failed to restore a viable, self-sustaining population (*Tissier et al., 2019*). Given the high mortality rates following release, it would seem obvious that efforts should focus on increasing the post-release survival of captive-bred hamsters to improve the effectiveness of this conservation measure.

In an effort to reduce post-release mortality of captive-bred hamsters, we investigated whether a pre-release confinement inside a semi-natural environment might elicit a more appropriate anti-predation response. To evaluate the efficacy of such confinement, hamsters were confronted with a predator model before and after a 2-week period inside a large field enclosure (field group) and their behaviour was recorded during these tests. To study whether potential behavioural differences after such confinement persisted over time, hamsters were confronted again with the same predator model 1 month after their return to the captive indoor facility. The objective of the confrontation with the predator

model was not to condition hamsters to react more appropriately to the risk of predation (*i.e.*, not to conduct an antipredator training), but to assess how a short period in a semi-natural environment might modify their anti-predator behaviour and elicit a more appropriate response (*Tissier et al., 2019*). In parallel, we investigated whether the repeated confrontation with a predator model alone could alter the behavioural responses in a group of hamsters that remained inside the captive facility throughout experimentation (control group). This experimental design allowed to address the following questions: (1) Does a 2-week confinement within a large field enclosure affect the anti-predation responses of 'naive' hamsters (confrontation test #1 *vs* #2)? (2) If anti-predation responses differ after the confinement, do these modifications persist over time (confrontation test #3)? And (3) Does repeated exposure to a predator model alone (without confinement period inside the field enclosure) alter the anti-predator behaviour of hamsters (Control group, confrontation test #1 *vs* #2)?

We expected that (1) a 2-week confinement in a large field enclosure will lead to a shift in the anti-predator behaviour of hamsters between confrontation trials, leading to a more appropriate response (*i.e.*, flee and hide rather than mounting an aggressive defense; hypothesis 1); that (2) any potential differences in anti-predator behaviour in the field group will diminish over time (hypothesis 2); finally, we expected that (3) hamsters without such confinement will not show a shift in their behavioural response between confrontation tests (hypothesis 3).

## MATERIALS AND METHODS

### Ethical note

This study followed the EU Directive 2010/63/EU guidelines for experiments, care and use of laboratory animals. The experimental protocol was approved by an Institutional Review Board (Ethical Committee: CREMEAS) under agreement number 02015033110486252 (APAFIS#397)02. At the end of the study, hamsters were not euthanized as they were only subjected to behavioural tests without invasive treatments. Individuals were released into the wild the same year or the following year as part of the annual reinforcement program.

### Animals and housing conditions

We used 27 1-year-old female European hamsters that were born and raised in our captive breeding unit (CNRS, IPHC-DEPE, Strasbourg, France). Only females were included in the study to (1) minimize competition and potential conflicts within the enclosure (since females have smaller territories and are less competitive than males) and (2) avoid reproduction within the enclosure (the behaviour of pregnant females likely differs from that of non-pregnant females, potentially adding confounding factors). Hamsters in this unit are the descendants of wild hamsters that were caught in the region (near Blaesheim, Alsace, France) between 1996 and 2002 (*Reiners et al., 2014*). After weaning, all hamsters were equipped with RFID tags (Radio-Frequency Identification 1.4 * 8.5 mm transponder), injected under the skin (Yes Mini, SAPV 32500; Groupe SNVEL, Paris, France), for permanent identification. Animals were housed individually in transparent Plexiglas cages with wire lids (420 * 265 * 180 mm, L * W * H) that contained bedding material and

enrichments (wood and shredded paper). Water and food pellets (105 pellets, SAFE, Augy, France) were provided *ad libitum*. During experimentation room temperature was maintained at 20–23 °C and light conditions followed the summer photoperiod (16L:8D). Hamsters were randomly assigned to two groups (control *vs* field group) before experimentation.

## Experimental design

To study the instinctive anti-predator responses of captive-bred hamsters and to investigate whether a confinement in the enclosure might suffice to change their anti-predator behaviour, we developed a standardized confrontation test with a predator model (Fig. 1). All experimental trials were conducted on the CNRS Campus (Strasbourg, France) during daylight hours (9 am–5 pm) and were filmed with a digital video camera. While hamsters are typically most active during dusk and dawn, we took advantage of the observation that hamsters in our breeding unit are also active during the day, when animal care staff cleans cages and provides food and water. All hamsters (N = 27) were raised under similar conditions prior to treatment. Before the confrontation tests, hamsters were allowed to familiarize themselves with the experimental arena during two habituation sessions (~12 min each), which were separated by 1 week. Following these sessions, all hamsters participated in two standardized confrontation tests with a predator model, which were separated by 2 weeks (Table 1). During these 2 weeks, hamsters underwent two different treatments; hamsters of one group were placed inside a large field enclosure (field group; N = 15), while hamsters of the other group (control group; N = 12) remained in their individual cages at the breeding facility. Sample size was greater in the field group to balance the potential loss of individuals during the period in the field enclosure (*i.e.*, escape/mortality from natural causes). A total of 2 weeks after the release of the field group into the enclosure, hamsters were recaptured by trap (MCL Leclercq, Wavrin, France) and underwent a second confrontation test 24 h after their return to the breeding unit. Similarly, control hamsters underwent their second test 2 weeks after their first, albeit without a confinement period in the field enclosure between tests (Table 1). To ensure a similar treatment between groups, hamsters of the control group were placed in individual wooden boxes and taken on a 25 km drive for 30 min the day before their second confrontation test. Hence, both groups experienced a disturbance related to transportation just before their second test round. Finally, 1 month after the second confrontation test, hamsters from the field group underwent a third test. The control group could not undergo a third test because these animals had already been released as part of the annual reinforcement program. However, given that the control group did not exhibit behavioural modifications between test 1 and test 2, we considered a third test, investigating the persistence of behavioural modifications, as unnecessary.

## Field enclosure, release and recapture protocol

The semi-natural environment, into which we released 15 hamsters of the field group, consisted of a 2,000 m² field enclosure, located near Blaesheim (Alsace, France, 48°30′ 14.044″N 7°36′28.414″E; elevation: 154 m above mean sea level). The vegetation inside the

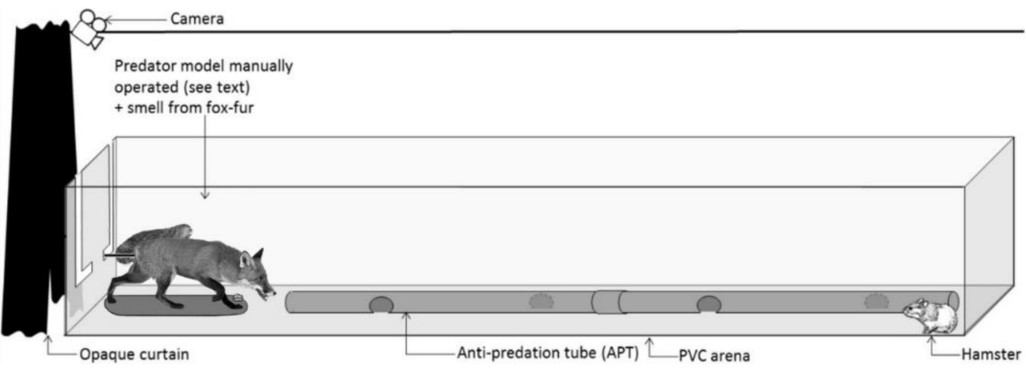

**Figure 1 Standardized test setup.** Representation of the PVC arena and the predator model. During the tests, a hamster was confronted with the predator model for 4 min. An experimenter controlled the movements of the fox model from behind the opaque curtain in response to hamster behaviour (see Fig. S1).                                                                    

**Table 1 Outline of experimental trials.**

|  | Before treatment (test #1) | Treatment | 24 h after treatment (test #2) | 1 month after treatment (test #3) |
|---|---|---|---|---|
| **Control group (N = 12)** | First confrontation test | 2 weeks in the laboratory | Second confrontation test | NA |
| **Field group (N = 10)** | First confrontation test | Confinement period (2 weeks in the field enclosure) | Second confrontation test | Third confrontation test (N = 9) |

enclosure consisted of a diverse mix of plants including, but not limited to, alfalfa (*Medicago sativa*), clover (*Trifolium pratense*), goldenrod (*Solidago virgaurea*), and peas (*Pisum sativum*). To prevent the intrusion of terrestrial predators, the entire area was surrounded by a chain-link fence (19 mm mesh size; vertical: 1 m above ground to 0.8 m below ground), reinforced with a 6 mm wire mesh. The latter was extended horizontally along the ground (1 m) towards the inside of the enclosure, to prevent hamsters from digging close to the fence to escape. To further prevent hamsters from escaping by climbing the fence, a galvanized metal sheet (20 cm high) was fixed to the inside of the fence. The top of the enclosure was covered by a net (mesh size: 50 mm) to exclude avian predators. As an additional measure against terrestrial predators, such as foxes, a single wire electrical fence was attached to the outside of the fence. Artificial burrows, consisting of a vertical and a sloping tunnel that met at ~1 m underground (*Müskens, Van Kats & Kuiters, 2008*), were created before hamster release. Within 24 h of the first confrontation tests, hamsters of the field group were released inside the field enclosure around sunset. Upon release, each hamster was placed into an artificial burrow. For 2 weeks, hamsters were free to explore the enclosure, to forage, to dig their own burrows and to interact with congeners and/or other small fauna. Food pellets were placed inside these burrows and vegetation was used to partially block the exits to reduce potential stress and to motivate hamsters to explore their burrow. Camera-Traps (Hyperfire HC600; Reconyx, Holmen, WI, USA), placed inside the enclosure, showed the presence of field mice (*Apodemus*

*sylvaticus*), shrews (*Crocidura leucodon*), rats (*Rattus rattus*), and field voles (*Microtus arvalis*). We found no evidence for the presence of potential predators inside the enclosure during the 2 weeks of treatment. However, foxes (*Vulpes vulpes*), buzzards (*Buteo sp.*), and ferrets (*Mustela putorius*) were observed in the vicinity of the enclosure. Naturally available food was supplemented daily with apples, onions, watermelon, and water. Food supplements were placed inside inactivated traps, to familiarize hamsters with these traps and, hence, improve the chances for a rapid recapture at the end of the 2-week period. During recapture, traps were activated at sunset and hamsters that were caught during the night were returned to the laboratory at sunrise. After the 2-week period, we recaptured 10 of the original 15 hamsters within one night. Another hamster was recaptured later (4 weeks after release) and was excluded from the study. Four hamsters were not recaptured and had likely escaped, despite our efforts to prevent this. This explanation is supported by multiple holes we found adjacent to the outside of the fence, which were likely part of tunnels passing underneath the fence. While it is also possible that agonistic interactions between hamsters occurred, potentially leading to the death of some individuals, we found no evidence of this (*i.e.*, no injured or dead hamsters were recorded and no agonistic interactions were visible in the pictures taken by camera-traps).

## EXPERIMENTAL PROTOCOL

### The arena and the predator model

All confrontation tests were conducted within a rectangular arena, constructed from PVC boards (3 * 1 * 0.4 m, L * W * H; Fig. 1). A PVC tube (2 m long, 10 cm in diameter), perforated at 50 cm intervals, was placed in the middle of the arena. It mimicked the shape of a tunnel, providing shelter, and was accessible to hamsters throughout a trial.

The efficacy of such a PVC tunnel to act as an 'anti-predation tube' (APT) had been confirmed previously during confrontation trials between hamsters and a mobile predator (the European ferret; *Tissier et al., 2016, 2018, 2019*). Foxes are one of the main predators of hamsters in the wild (*La Haye et al., 2020*). Hence, we used a taxidermically-mounted red fox as a predator model in confrontation trials. The fox was mounted in an attack posture (open mouth showing teeth, a curved back, and the tail pointing upwards, Fig. 1).

To increase the realistic depiction of the model, we also presented fresh fox scent at the beginning of each test. The source of this scent was hair collected from eight non-sterilized adult foxes (four males and four females) at the Nancy Laboratory for Rabies and Wildlife (ANSES, Malzéville, France). To ensure a similar olfactory stimulation during trials and to prevent the accumulation of scent on the fur of the mounted fox, the collected hair was contained in a small plastic container, positioned between the two front legs of the fox model. At the end of each trial, the hair sample was removed, and the container thoroughly cleaned (70% ethanol) to remove all scent. In addition, the PVC arena was cleaned with ethanol (70%) and the room was aired out for 10 min between trials.

### Confrontation tests

The confrontation test was a standardized behavioural test to assess the behaviour of hamsters before and after the treatment. These tests were not a training measure to

condition the anti-predator behaviour of hamsters. Each test lasted 14 min and was divided into three phases. During an initial 5-min period, a hamster could move freely inside the PVC arena without external perturbation (phase 1). Thereafter, the fox model and associated scent were presented to the hamster for 4 min (phase 2; Fig. 1). Finally, the fox model and associated scent were removed from the arena and the hamster was left undisturbed again for 5 min (phase 3). During the confrontation with the fox model (phase 2), an experimenter was hidden behind an opaque curtain and controlled the movements of the fox model *via* a metal rod fixed below the tail of the fox (Fig. 1), mimicking predator attacks. The experimenter followed a strict protocol, adapted to the behaviour of the hamster (Fig. S1).

## Behavioural recordings and statistical analysis

All confrontation tests were filmed, and video analysis was conducted using the *Behavioral Observation Research Interactive Software* (Boris, v.6.3.3-2018; *Friard & Gamba, 2016*). The identity of a hamster (*i.e.*, whether it belonged to the field or control group) during the confrontation tests and subsequent video analyses, was unknown to the experimenter. At the start of the video analysis an ethogram, containing various behavioural variables, was established (Table S1). The start and end times of different behaviours included in the ethogram were marked and durations summed to establish a time budget for the different behavioural variables. We focused our analysis on the following four behavioural variables (see Table S1): (i) the time (% of phase duration) the hamster spent inside the shelter (APT); (ii) the time (fraction) the hamster spent exploring the arena when outside the APT; (iii) the time between introduction of the fox model into the arena and entrance of the hamster into the APT (latency); (iv) the number of hamster attacks on the fox model. The first two variables were investigated separately for each trial phase, while the last two variables only concerned phase 2.

To test (1) whether a 2-week confinement into a large field enclosure was sufficient to shift the anti-predator behaviour of hamsters (field group) between tests, leading to a more appropriate response (hypothesis 1) and to also test (2) whether such a shift was absent in hamsters without such confinement (control group; hypothesis 3), we ran Generalized Linear Mixed Models (GLMMs) for each of the four behavioural response variables, comparing tests #1 with tests #2 (before/after treatment) of control and field hamsters (Table S2). Group and test number were included as fixed effects, while Hamster ID was included as a random effect to account for repeated measures. Interactions between group and test number were also included (*e.g.*: Behavioural variable = group + test# + group * test# + hamster ID (random)). We used an ANOVA based GLMM with Tuckey-HSD for *post-hoc* analyses.

Similarly, to test if potential differences in anti-predator behaviour in the field group diminished over time (hypothesis 2), we ran GLMMs for each of the four behavioural response variables and tested for differences across their test numbers (#1 to #3). We used an ANOVA based GLMM with Tuckey-HSD for *post-hoc* analyses to compare field group tests numbers. The same procedure was also used to test for potential differences between groups during the first test round.

All analyses were conducted in R (v3.5.1; *R Core Team, 2022*) with the RStudio interface (RStudion version 1.3.959; *RStudio Team, 2020*), using the following packages: "tidyverse", "lme4", "MASS", "multcomp", "car", "ggpubr" and "nlme". Figures were plotted using GraphPad prism software (v9.0.1; GraphPad Software, San Diego, CA, USA).
The significance threshold was set at $p < 0.05$. All values presented are grand means ± s.e., established from individual hamster means, unless specified differently.

## RESULTS

We conducted a total of 58 confrontation tests (control group: 24 tests; field group: 34 tests). While the latter group originally consisted of 15 individuals, only 10 individuals were recaptured from the enclosure at the end of the confinement period. In addition, one hamster of the field group died for unknown reasons 2 weeks after recapture and, hence, could not be tested during the last round. Accordingly, all data concerning the missing five individuals of the field group were removed from the analysis, leaving a final sample size of $n = 29$ tests for field hamsters ($n = 10$ for both test #1 and #2, and $n = 9$ for test #3).

### Behavioural differences between trials

Comparing the behavioural variables displayed during tests #1 and #2 showed overall no significant differences between groups or tests (Table S3). However, the interaction term between groups and test number was significant, indicating that the test comparison differed between groups (Table S3). *Post-hoc* analysis, comparing tests for each group separately, showed that most of the behavioural variables differed significantly between test #1 and test #2 in the field group, but not in the control group (Table 2 and Figs. 2–4). For example, hamsters of the field group spent a significantly greater proportion of time hiding inside the PVC tube during and after predator confrontation following the confinement period (+34% and +51% during phase 2 and 3, respectively) than during the same phases in test #1 (Table 2 and Fig. 2). Similarly, hamsters of the field group spent a significantly smaller fraction of time exploring the arena before and after predator confrontation following the confinement period (−20% and −16% during phase 1 and phase 3, respectively; Table 2 and Fig. 3) The latency period before field hamsters entered the APT was, on average, greatly reduced following the confinement period (87.5 ± 95.6 s *vs* 3.6 ± 4.6 s before/after confinement period, respectively), albeit, due to individual variation, this difference was not significant ($p = 0.09$; Table 2 and Fig. 4A). Finally, the number of attacks by hamsters on the fox-model decreased significantly after the confinement period in the field group (on average 9.6 and 3.9 attacks before/after confinement period, respectively, −60%; Table 2 and Fig. 4B).

To ensure that behavioural differences did not exist between groups before the confinement period, we compared the behaviour of both groups during their first confrontation test. Our analysis did not find significant differences between groups during test #1 for the behavioural variables studied, with one exception: during the confrontation phase, hamsters of the field group spent less time hiding inside the tube than control hamsters ($p = 0.003$; Table S5).

**Table 2 Model results (*post-hoc* tests) comparing hamster behaviour of the control and field group during tests 1 and 2 (before/after treatment) according to test phase.**

| | Variable | Phase | Estimate ± SE | Z | Df | p | Behavioural differences between test 1 and test 2 |
|---|---|---|---|---|---|---|---|
| **Control group** | Time (%) spent inside APT | 1 | 0.91 ± 0.36 | 2.55 | 1 | 0.052 | ↘ |
| | | 2 | 1.04 ± 0.53 | 1.97 | 1 | 0.187 | – |
| | | 3 | 0.82 ± 0.47 | 1.74 | 1 | 0.3 | ↘ |
| | Exploration (%) when outside APT | 1 | 0.09 ± 0.22 | 0.41 | 1 | 0.976 | – |
| | | 2 | NA | NA | NA | NA | NA |
| | | 3 | 0.01 ± 0.39 | 0.02 | 1 | 1 | – |
| | Latency before first entry into APT | 2 | 0.18 ± 0.08 | 2.21 | 1 | 0.095 | ↗ |
| | Attacks on fox model | 2 | −0.83 ± 0.54 | −1.52 | 1 | 0.377 | – |
| **Field group** | Time (%) spent in the APT | 1 | −0.82 ± 0.36 | −2.31 | 1 | 0.093 | ↗ |
| | | 2 | −2.43 ± 0.57 | −4.30 | 1 | <0.001 | ↗↗↗ |
| | | 3 | 2.33 ± 0.59 | −3.94 | 1 | <0.001 | ↗↗↗ |
| | Exploration (%) when outside APT | 1 | 0.85 ± 0.24 | 3.47 | 1 | 0.003 | ↘↘ |
| | | 2 | NA | NA | NA | NA | NA |
| | | 3 | 1.62 ± 0.57 | 2.82 | 1 | 0.023 | ↘ |
| | Latency before first entry into APT | 2 | −0.27 ± 0.12 | −2.24 | 1 | 0.087 | ↘ |
| | Attacks on fox model | 2 | 0.90 ± 0.23 | 3.95 | 1 | <0.001 | ↘↘↘ |

**Note:**
Bold arrows indicate the direction of a significant difference (increase/decrease), while plain arrows indicate only a (non-significant) trend and hyphens indicate no change between tests. The number of arrows indicates if the difference is less than or equal to 0.05 (one arrow), 0.01 (two arrows) or 0.001 (three arrows). When under attack (phase 2), hamsters never explored the arena, as indicated by NA.

### Persistence of behavioural changes over time

For field group hamsters, the behavioural variables during test #2 (after confinement period) did not differ from those of test #3 (1 month after the return to the laboratory; Table 3), indicating that behavioural modifications following the 2 weeks inside the enclosure persisted for at least 1 month. However, if we also include test #1 in such investigation, we find that some behavioural modifications started to revert between test #2 and #3, so that they did not differ significantly from test #1. This concerned for example the time spent inside the APT during phase 3 (Fig. 5) or the time spent exploring during phase 3 (Fig. S2). However, most behavioural modifications persisted over time (Table S4 and Fig. S3).

### DISCUSSION

We found that a 2-week confinement inside a large enclosure was sufficient to elicit significant changes in the behavioural responses of captive-bred European hamsters when confronted with a predator model. Following their period in the enclosure, these hamsters showed a response to a predator model that is likely more appropriate when encountering a predator (*i.e.*, hiding/fleeing rather than attacking). After the confinement, hamsters of the field group spent more time within the APT providing shelter from the predator model, spent less time exploring the arena before and after predator confrontation, and attacked the predator model less frequently (Table 2 and Figs. 2–4). By contrast, repeated

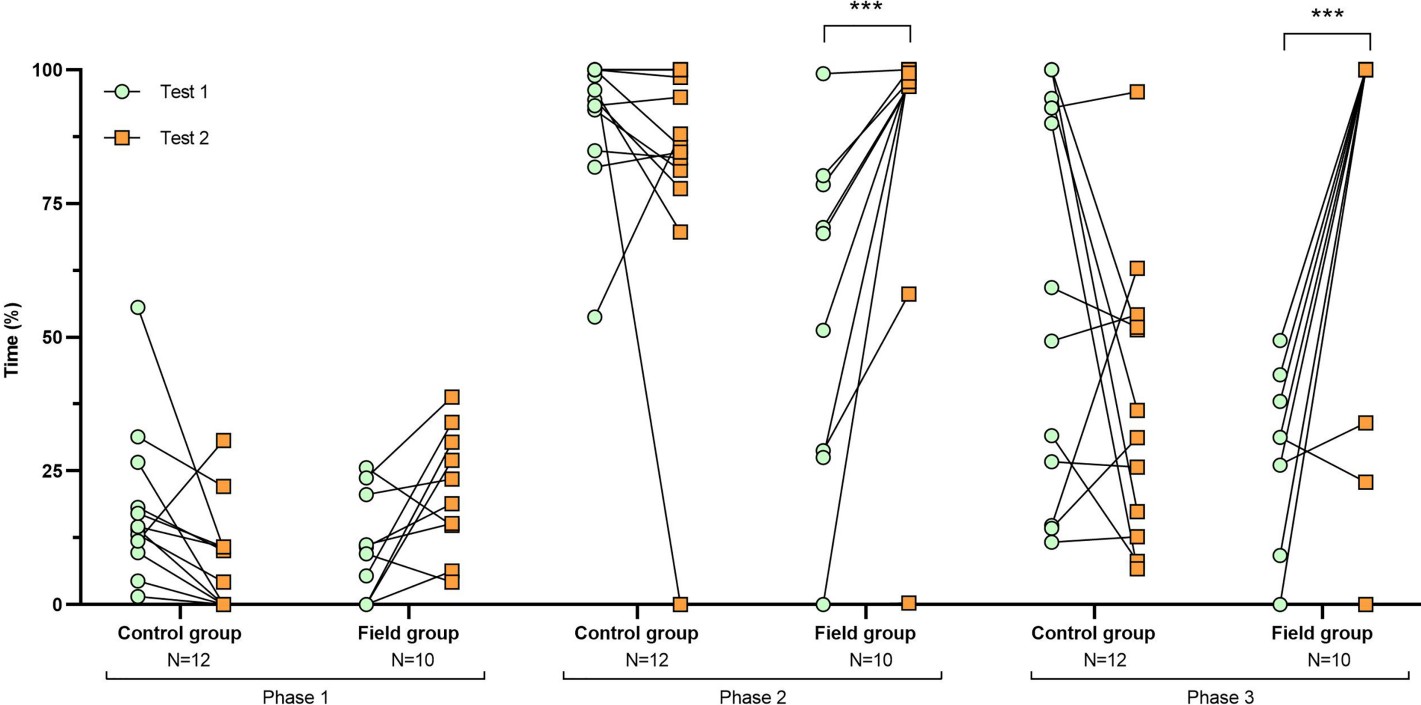

**Figure 2 Time spent inside the APT during tests before and after treatment.** Values for individuals are presented (circles/squares) according to group (control/field), test number (test #1: before treatment; test #2: after treatment), and phase during a trial (phase 1–3, before, during, and after predator confrontation, respectively). Asterisks indicate significant differences between test numbers (*** ≤0.001).

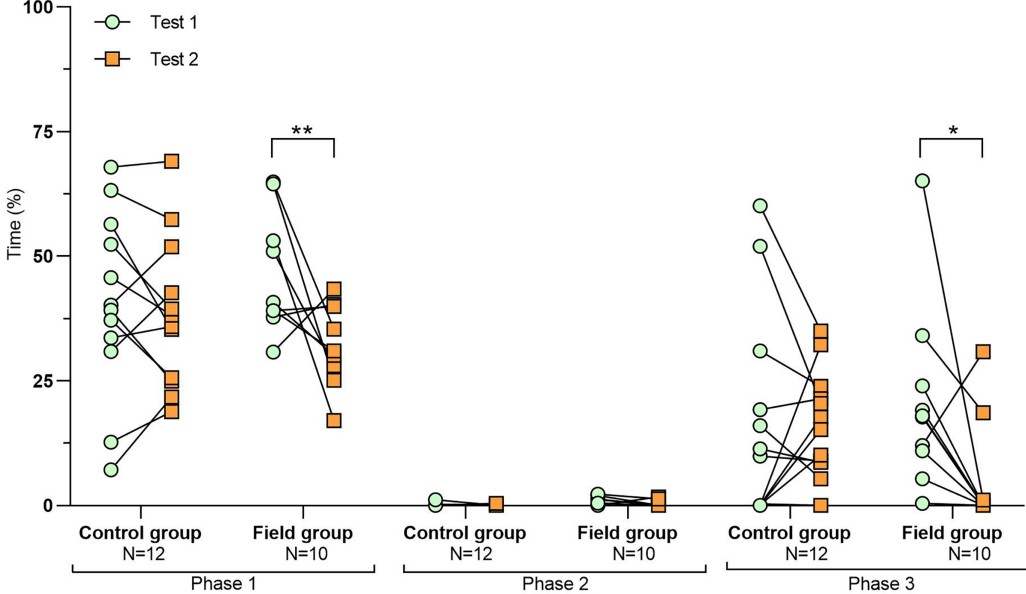

**Figure 3 Time spent exploring the arena during tests before and after treatment.** Values for individuals are presented (circles/squares) according to group (control/field), test number (test#1: before treatment; test #2: after treatment), and phase during a trial (phase 1–3, before, during, and after predator confrontation, respectively). Asterisks indicate significant differences between test numbers (* ≤0.05 and ** ≤0.01).

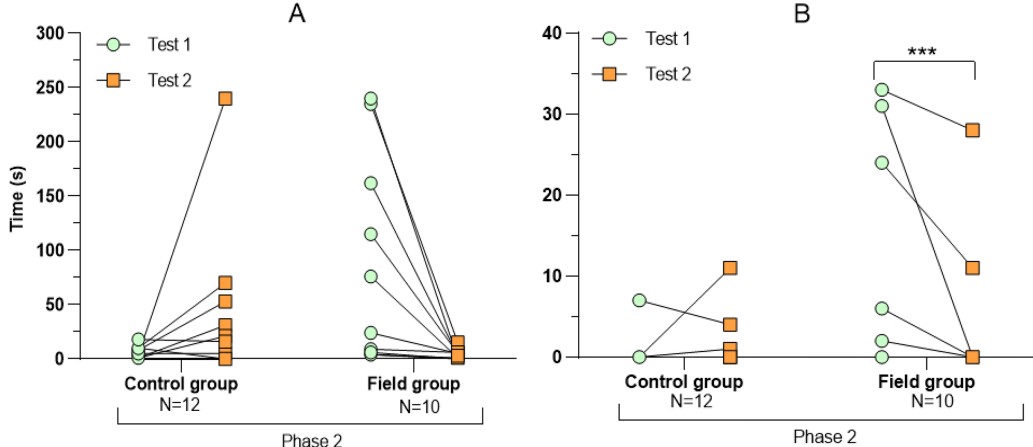

**Figure 4 Latency period (A) and number of attacks on the predator (B) during tests before and after treatment.** Values for individuals are presented (circles/squares) according to group (control/field) and test number (test #1: before treatment; test #2: after treatment). Asterisks indicate significant differences between tests (*** ≤0.001).

**Table 3 Model results (*post-hoc* tests) for behavioural variables of field group hamsters during test 2 (immediately following treatment) and 3 (1 month after treatment).**

|  | Variables | Phase | Estimate ± SE | Z | Df | p |
|---|---|---|---|---|---|---|
| Test 2 *vs* Test 3 | Time (%) spent inside APT | 1 | −0.22 ± 0.45 | −0.48 | 2 | 0.88 |
|  |  | 2 | −0.28 ± 0.62 | −0.44 | 2 | 0.90 |
|  |  | 3 | −0.91 ± 0.68 | −1.34 | 2 | 0.37 |
|  | Exploration (%) when outside APT | 1 | −2.25 ± 0.36 | −0.70 | 2 | 0.76 |
|  |  | 2 | NA | NA | NA | NA |
|  |  | 3 | 0.82 ± 0.66 | 1.23 | 2 | 0.43 |
|  | Latency before first entry into APT | 2 | −0.23 ± 0.12 | −1.92 | 2 | 0.11 |
|  | Attacks on fox model | 2 | −0.77 ± 0.76 | −1.02 | 2 | 0.56 |

**Note:**
When under attack (phase 2), hamsters never explored the arena, as indicated by NA.

exposure to the predator model alone, without a confinement period inside the enclosure (control group), was not sufficient to elicit significant changes in hamster behaviour during trials (Table 2 and Figs. 2–4). The behavioural modifications following treatment in the field group persisted over time (at least 1 month; Table 3) but a partial reversal was noticeable for some behavioural variables (Figs. 5 and S2).

## Behavioural modifications following confinement in the field enclosure

Following the period spent inside the field enclosure, field hamsters showed significant changes in their anti-predator behaviour during confrontation trials. In addition, since in our experimental design individuals served as their own control, we could evaluate the effects of the confinement period on the behaviour of individuals (*i.e.*, field *vs* control group).

During the first test-round, hamsters of the field group spent significantly less time hiding inside the APT during and after predator exposure and also mounted a greater

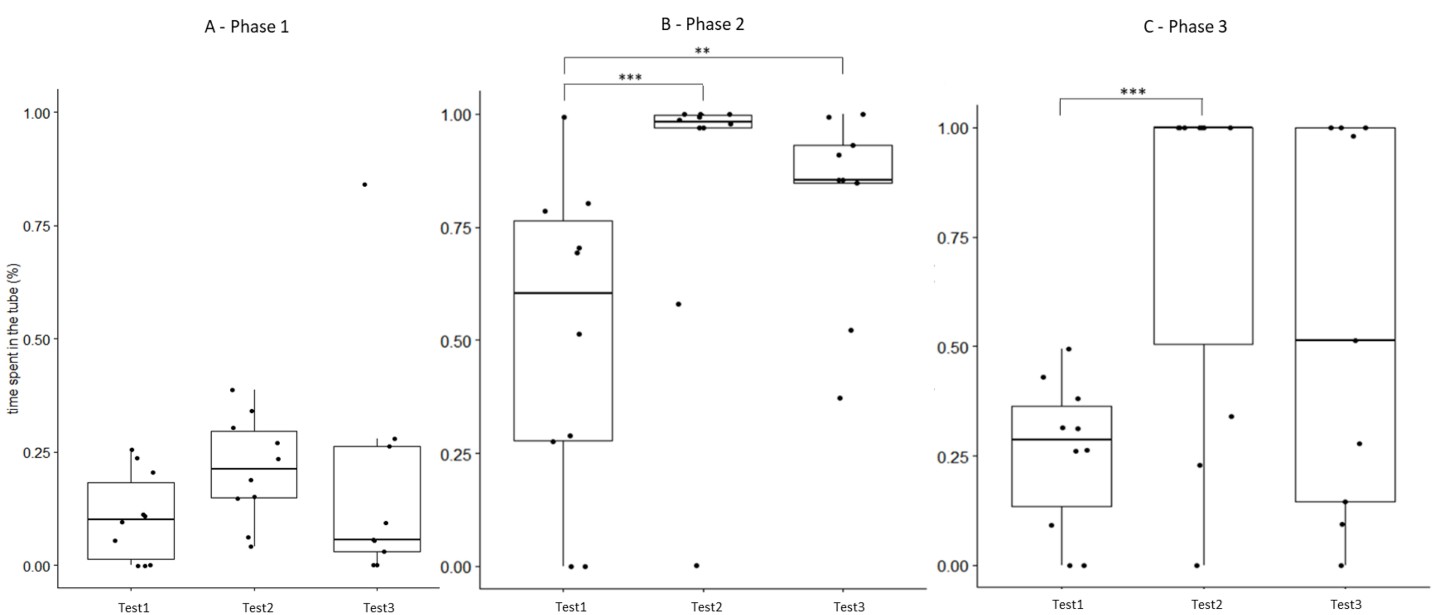

**Figure 5 Time that field hamsters spent inside the APT during three confrontation tests according to phase.** Test #1 (pre-confinement), test #2 (post-confinement), and test #3 (1-month after confinement) according to test phase (before, during, and after predator model exposure). Individuals are indicated by black dots and significant differences between tests are indicated by asterisks (** ≤0.01 and *** ≤0.001).

number of attacks on the fox model than control hamsters (Figs. 2 and 4). Hence, field hamsters originally displayed an anti-predator response that could be characterized as more risk-prone than that of control hamsters. Hence, despite randomizing the individual assignment to groups, it is likely that a greater number of bold hamsters (maintaining a high activity level during predator presence; *Watters & Meehan, 2007*) were assigned to the field group (Fig. 4). However, after the confinement, the behavioural response of hamsters from the field group changed significantly (Table 2 and Figs. 2–4). After the confinement, these hamsters spent significantly more time inside the shelter and reduced the latency period before entering the shelter during predator model confrontation. They also significantly decreased the time exploring the arena before and after predator confrontation and reduced the number of attacks on the fox model. This suggests that their "risk-prone" behavioural response during the first test shifted to a 'risk-averse' response following treatment.

The confinement in the field enclosure was intended to provide hamsters with the possibility to learn from a protected semi-natural environment, rather than to expose them to a real predator-prey confrontation. To the best of our knowledge, measures to avoid predation worked effectively. All hamsters in our study had been housed in individual cages since weaning, and they never encountered conspecifics within their 'habitat'. During the confinement, hamsters of the field group were exposed to a multitude of new stimuli. For the first time, they experienced natural climatic conditions, a natural soundscape, other animals, as well as intraspecific competition. They were able to express behaviours they could not engage in while inside the laboratory housing (*e.g.,* digging a burrow, foraging, exploring a large area, *etc.*). Hence, beyond the possibility to learn from the

protected exposure to predation risk, field hamster experienced a variety of stimuli (*e.g.*, tactile, olfactory, sound) that differed greatly from those in the laboratory. This should have enabled hamsters to develop their cognitive and behavioural capacities and to adapt their digestive and immune systems to a more natural environment (*Shepherdson, 1994*; *Salvanes et al., 2013*). The development of such capacities is strongly dependent on rearing conditions and the immediate experience before the release (*Reading, Miller & Shepherdson, 2013*; *Tetzlaff, Sperry & DeGregorio, 2019*). The behavioural modifications following the confinement suggest that hamsters learned to associate certain stimuli (*e.g.*, unknown smells or sounds) with a potential risk, triggering an appropriate reaction (*e.g.*, seeking shelter, being more vigilant in open spaces).

One might argue that the potential stress associated with a relatively high density of animals in the enclosure and/or the recapture and return to the captive facility just before the second test are responsible for the observed behavioural modifications of field hamsters. However, it is important to distinguish between 'chronic distress' and 'natural stress' (*Reading, Miller & Shepherdson, 2013*). The first may impact captive-bred animals and may lead to the development of abnormal behaviour (*e.g.*, pacing in a cage, pulling out fur) because animals lack the adaptive behavioural outlet to control their situation. By contrast, 'natural stress' may occur sporadically/periodically and is necessary for the development of adaptive psychological and behavioural skills (*Moodie & Chamove, 1990*; *Meehan & Mench, 2007*; *Reading, Miller & Shepherdson, 2013*). In this context, the limited exposure to stressful stimuli inside the field enclosure might have been advantageous. Nevertheless, hamster density within the enclosure was ~10 times greater than what is typically found in the wild (apart from very high densities that occur during population outbreaks). While this may have potentially added some stress to the hamsters in the enclosure, a high density might be particularly challenging when resources, such as food or access to partners, are limited, which was not the case in our situation. If such potential additional stress would have altered the behavioural responses of hamsters during the test that followed treatment, one would expect consistent changes throughout the different test periods (*e.g.*, hiding inside the APT throughout a test). However, this was not the case (*e.g.*, hamsters were hiding inside the APT during and following predator exposure but not before). Hence, there is little evidence that a potentially increased stress level during treatment in the field hamsters may have been responsible for the behavioural modifications observed during tests #2 and #3.

Captive-bred animals, especially in a research laboratory, lack sufficient stimulation from external factors (*e.g.*, predators, congeners, natural soundscape, weather) that would enable them to develop behavioural responses more appropriate for a natural environment (*Mathews et al., 2005*; *Wells, 2009*; *Salvanes et al., 2013*). In addition, the cramped conditions of the standard breeding cages for rodents likely contribute heavily to the inferior physical, neuro-motor, psychological, and sensory conditions of captive rodents (*Young, 2003*). Cognitive processes are essential for mounting the appropriate behavioural response in a given situation (*Curio, 1993*; *Griffin, Blumstein & Evans, 2000*). The switch from a fight to a flight response that we observed during confrontation tests with field hamsters following their return from the enclosure, suggests that their confinement period

improved cognitive processes, triggering more appropriate behavioural responses to predation risk. Hence, we suggest that a pre-release preparation period inside a field enclosure, where hamsters are exposed to a variety of novel stimuli, will likely lead to an improvement of their overall condition and will be an important measure to reduce mortality of hamsters.

## Does repeated predator model confrontation alone elicit behavioural changes?

Hamsters of the control group, which remained within their standardized cages between the two tests, did not display significant changes in their behavioural responses between test #1 and test #2 (Table 2 and Figs. 2–4). Hence, repeated confrontation with the predator model and its scent alone, was insufficient to elicit a more appropriate anti-predation response (*e.g.*, avoidance, shelter seeking). Even the multiple direct attacks by the predator model during trials, that involved physical contact and that, under natural conditions, would have resulted in death by predation, did not suffice to provoke changes in anti-predatory responses.

For survival, prey species must first detect a potential predator and then react appropriately (*Lönnstedt et al., 2012*; *Blumstein, Letnic & Moseby, 2019*). For this, however, they first have to be able to recognize a predator as potential danger (*McLean, Lundie-Jenkins & Jarman, 1996*). All hamsters tested in this experiment reacted to the exposure and attacks of the fox model in all tests (*i.e.*, suppression of exploration, increased use of the shelter, attacks on the fox) and, hence, likely perceived the fox model as potential danger (Table 3 and Fig. 4). However, while field hamsters also modified their behaviour during phase 1 and 3 (before/after confrontation) following treatment (*e.g.*, increased use of shelter, reduced exploration) this was not the case for control hamsters (Figs. 2–3). Training/conditioning captive-breed animals to recognise their natural predators has been attempted with many species, albeit with varying success (*Vilhunen, 2006*; *Lönnstedt et al., 2012*; *Lopes et al., 2017*; reviewed in *Rowell, Magrath & Magrath, 2020*). For some species, simple exposure to predator odours was sufficient to increase their survival during a following predator confrontation (*Vilhunen, 2006*). By contrast, multiple confrontations with a predator model in association with aversive stimuli were insufficient to improve the post-release survival of parrots (*Lopes et al., 2017*). In addition, such a method, where captive animals are repeatedly exposed to an (artificial) predator model or to a risk of predation under controlled conditions might be counterproductive, as it could lead to habituation (*Rowell, Magrath & Magrath, 2020*). For example, anti-predator behavioural responses might diminish over time, due to habituation to the threat and/or due to learning of inappropriate responses to a predation threat (*Rowell, Magrath & Magrath, 2020*; *Edwards et al., 2021*). Furthermore, even if a live predator is used, it might be difficult to reproduce the exact stimuli that animals experience during a predator encounter in the wild (*Griffin, Blumstein & Evans, 2000*). In our experiment, the number of confrontation trials and exposure duration to the predator model that we used were likely insufficient to elicit any habituation. In this context, it would be interesting to investigate if and how a

longer exposure to a predator model and/or a greater number of trials, without a confinement in a field enclosure, affects the anti-predator responses of naïve hamsters.

## Persistence of behavioural changes over time

The behaviour of field hamsters displayed during tests #2 and #3 did not differ significantly (Table 3), suggesting that the behavioural modifications after the confinement persisted over time and were still present 1 month after their return to the captive facility. For example, during confrontation, field hamsters spent significantly more time inside the APT following the confinement and also 1 month thereafter (test #2 and #3, respectively), when compared with test #1 (Fig. 5). Similarly, the number of attacks on the predator model by field hamsters was reduced following the confinement and remained at such level during test #3. However, when including test #1 in our analyses, we found that some behavioural modifications of field hamsters started to revert between test #2 and #3 and did not differ significantly anymore from test #1. This was the case, for example, for the time spent inside the APT during phase 3 (Fig. 5) or the time spent exploring during phase 3 (Fig. S2). However, most behavioural modifications persisted across tests. Nevertheless, since the intensity of the behavioural modifications had started to fade 1 month after the confinement period, further reinforcements might be required for behavioural modifications to persist. The ability of animals to retain behaviours acquired during predator-awareness training (*i.e.*, *via* conditioning) have been studied in a variety of animals (*McLean, Lundie-Jenkins & Jarman, 1996*; *Griffin, Blumstein & Evans, 2000*; *Rowell, Magrath & Magrath, 2020*). Depending on the training regime and the species in question, anti-predator behaviours acquired during such training/conditioning have been shown to persist for up to several months, even in the absence of subsequent reinforcements (*Chivers & Smith, 1994*; *De Azevedo & Young, 2006*). Hence, our results suggest that hamsters have the capacity to retain some modifications of their behavioural response for at least 1 month, even in the absence of reinforcements (*i.e.*, a further pre-release confinement in the field enclosure), while others might be more susceptible to reversal.

## CONCLUSIONS AND PERSPECTIVES FOR HAMSTER CONSERVATION

Our study shows that a simple pre-release confinement in a field enclosure was sufficient to elicit a shift in the behaviour of hamsters towards a more adapted anti-predation response, when confronted with a predator model. In addition, most of the observed behavioural changes were retained for at least 1 month. Hence, the confinement period that hamsters spent inside the field enclosure was critical to achieve behavioural modifications that will likely improve their survival when facing the risk of predation upon their release into the wild. The repeated exposure to a predator model alone was insufficient to provoke behavioural modifications (control group). Our findings have important implications for hamster reinforcement programs. We suggest that a confinement period inside a field enclosure (*i.e.*, a 'soft-release'), as implemented here, should be applied before any release into the wild. However, the greater variation we observed during the last test round

suggests that behavioural changes fade over time in the absence of reinforcements (Figs. 5 and S2). Consequently, a release into the wild should be implemented as soon as possible after the confinement inside the field enclosure. A 1-month persistence of behavioural modifications, as found here, might be sufficient to increase hamster survival chances during the most critical period following release.

Our experimental approach, testing the effectiveness of a confinement period spent inside a field enclosure to elicit a more adept anti-predator response of captive bred hamsters is only a first step. We now need to evaluate if the short-term survival of these hamsters after their release is indeed increased, when compared with hamsters that did not undergo a pre-release preparation program (*e.g.*, see *Shier & Owings, 2006*; *Greggor, Price & Shier, 2019*). In addition, to ensure the success of restocking programs, released hamsters do not only have to survive, they also have to reproduce and successfully wean offspring (*Soorae, 2018*). The latter is of particular importance for hamsters, given their short lifespan. Hence, survival and successful reproduction of captive-bred hamsters in the wild are key demographic factors to consider for meaningful conservation measures. In this context, additional studies investigating how the treatment of hamsters prior to their release into the wild affects their reproductive rate are of great importance. Restocking programs are an important instrument in biodiversity conservation and should, therefore, also consider the well-being of animals before, during, and after release (*Swaisgood, 2010*).

## ACKNOWLEDGEMENTS

We thank Frederic Voegel for his help and support. Solène Liegeois and Nicolas Durr helped with different aspects of the study. We also thank the ANSES laboratory ('Laboratoire de la rage et de la faune sauvage de Nancy') for providing the fox hair samples.

### Funding
This work was supported by the Centre National de la Recherche Scientifique et la Direction Régionale de l'Environnement, de l'Aménagement et du Logement Grand-Est. The funders had no role in study design, data collection and analysis, decision to publish, or preparation of the manuscript.

### Grant Disclosures
The following grant information was disclosed by the authors:
Centre National de la Recherche Scientifique et la Direction Régionale de l'Environnement. de l'Aménagement et du Logement Grand-Est.

### Competing Interests
Jonathan Jumeau is an employee of Collectivité européenne d'Alsace.

## Author Contributions

- Julie Fleitz conceived and designed the experiments, performed the experiments, analyzed the data, prepared figures and/or tables, authored or reviewed drafts of the article, and approved the final draft.
- Manfred R. Enstipp conceived and designed the experiments, authored or reviewed drafts of the article, and approved the final draft.
- Emilie Parent conceived and designed the experiments, performed the experiments, analyzed the data, prepared figures and/or tables, and approved the final draft.
- Jonathan Jumeau analyzed the data, authored or reviewed drafts of the article, and approved the final draft.
- Yves Handrich conceived and designed the experiments, authored or reviewed drafts of the article, and approved the final draft.
- Mathilde L. Tissier conceived and designed the experiments, analyzed the data, authored or reviewed drafts of the article, and approved the final draft.

## Animal Ethics

The following information was supplied relating to ethical approvals (*i.e.*, approving body and any reference numbers):

The study followed the EU Directive 2010/63/EU guidelines for experiments, care, and use of laboratory animals. The experimental protocol was approved by the Ethical Committee (CREMEAS) under agreement number 02015033110486252 (APAFIS#397)02.

## Data Availability

The raw data is available at OSF: Fleitz, Julie. 2023. "Soft-Release and Anti-Predator Behaviour." OSF. June 1. DOI 10.17605/OSF.IO/HFX8T.

## Supplemental Information

Supplemental information for this article can be found online at http://dx.doi.org/10.7717/peerj.15812#supplemental-information.

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
