# Peer review of "Improving the success of reinforcement programs: effects of a two-week confinement in a field enclosure on the anti-predator behaviour of captive-bred European hamsters"

_PeerJ, doi:10.7717/peerj.15812_

## Round 0.1 · original submission · Major Revisions

Please carefully address all of the reviewers' concerns. Two of the reviewers were very critical. Be aware that a revised version will be re-reviewed, probably by them, so you need to satisfy their serious reservations about your manuscript.

·

Basic reporting

This article describes an experiment to evaluate behaviour of captive bred (European) hamsters after a two-week training period in a large enclosure. The species is subject of several conservation and reintroduction projects across Europe, but so far these releases are not very successful. Increasing survival of released captive-bred hamsters has possible a positive effect on establishing a new (wild) population. The article describes in much detail how different test are done after a 'two-week training', including the effects in a control group and the treatment-group ('field group'). The control group did not show changes in behaviour during trials, but the 'field-group' did show some changes. However, it remains unclear why this group changed behaviour, rather than having spent some time in the outdoor-enclosure. As predators were excluded from the enclosure, it is unlogical that these hamsters have learned to behave more appropriate when confronted with a predator? However, if a more appropriate anti-predator behaviour has been adopted in the enclosure, these hamsters will probably have a better survival after release. Still, some important questions remain: the authors released 15 individuals in the enclosure and trapped only 10. What has happened to the other 5 individuals? To understand the effect and efficiency of this 'two-week' training it is very interesting to know what happens within the enclosure (and why). Unfortunately, the authors hardly describe this part of the experiment and focus on the tests and comparison of the behaviour during tests and different phases of both groups, while as a reader you want to read more about the 'training' and conditions in the enclosure. Last, the title is confusing, as it suggest that a soft-release method has been tested, i.c. a group of captive-bred hamsters is released in the wild in a 'soft-way', rather than in a 'hard way' and that the effect of this treatment will be evaluated.

Experimental design

Although the different experimental, confrontation test have been discussed at length and described in detail, which is done very well, the article gives hardly any information about the main experiment: the two-week training in the outdoor enclosure and the possible consequences of this training.

Validity of the findings

The authors describe their findings in a detailed way with much attention for the experiments with the stuffed predator, which is interesting on its own, but the authors do not describe or overlook the possible effect(s) of housing captive-bred hamsters in an overcrowded enclosure. Only when studying the supplemental documents, it become clear that 15 individuals have been released in a fairly small enclosure of 2000 m2, resulting in an unnatural high density of hamsters. Such a high density may result in chronic stress or even death of individuals, perhaps the reason why only 10 individuals were re-trapped? It is well known that hamsters can be very territorial and kill each other, but otherwise also can have very high densities during a population outbreak. Still, it seems quite stressful to have so many hamsters in an enclosure. A reflection by the authors if these hamsters may have experienced some extra stress or something else in the enclosure, which has an effect on the tests, is really missing and an important reason for deciscion to decline the article.

Additional comments

The authors have really nice results and have done wonderful experiments, but they have come up with a confusing title and should have focused more on possible changes in behaviour and persistence of this new behaviour, then trying to link the results to a more successful release of captive-bred hamsters in the wild.

·

Basic reporting

In general the article is well written, with few minor typos. Figures and tables are sufficient. Raw data is not available. The Discussion section often repeats data from Results.

Experimental design

This is the original research which meets aims and scope of the Journal.
Experiment design is sound and well presented.
Minor suggestions are included in the annotated PDF.
Authors follow ethical standards.

Validity of the findings

The findings are valid and important. The final recommendations of the authors could be game changers in rewilding of hamsters and potentially other threatened species.

Additional comments

No additional comments

Reviewer 3 ·

Basic reporting

This is a very interesting study with a very actual scope. The text is well written, though adding some content-related info in some sections is advised. Specific literature on similar trails with the European hamster could be elaborated. Raw data are shared, and figures and tables are present. Some figures mentioned in the text were not added to the supplementary material.

Experimental design

The experimental design has some weak points which could bias the results of this whole trial. Variables are added to the field group (catching/transport) without adding these to the control group which makes a comparison biased. The effect of catching/transport/adjustment from the outdoor enclosure to small breeding cages will affect the results from the field group in trail 2 since it was executed short after these actions.
Also nothing is said about the proportion male/females within each group which could have an effect on the overall results of behavior.
Moreover, the control group was only confronted with a fake predator in 2 trails while the field group 3 times so accurate comparison was not possible.
I would suggest to alter the experimental design to one where only 1 variable is tested as stated in your hypothesis: does a training in an outdoor inclosure after a confrontation test provide an European hamster a more appropriate anti-predator behavior.

Validity of the findings

The findings are in my opinion biased which makes some conclusions questionable. Although the scope of this research is very meaningful the testing of the hypothesis could be enhanced.

Additional comments

For more general comments and detailed per line see attached file.

Annotated reviews are not available for download in order to protect the identity of reviewers who chose to remain anonymous.

---

## Round 0.2 · accepted · Accept

Thank you for your revisions.

·

Basic reporting

The authors made a substantial improvement to the manuscript. I am happy with the article as is and believe it is ready to be published in PeerJ. The only minor suggestion would be to add the author and year of description to the Latin name of the species during the first mention (line 118), as regulated by the ICZN.

Experimental design

Original primary research within Aims and Scope of the journal

Validity of the findings

All underlying data have been provided; they are robust, statistically sound, & controlled.

Reviewer 3 ·

Basic reporting

More literature is added.

Experimental design

This revised document shows a more detailed overview on the study design which makes it more valid.

Validity of the findings

Considering the more detailed info of some aspects in this revised paper the validity of the results and the conclusion is more robust.